# Systematic Localization of *Escherichia coli* Membrane Proteins

Anna Sueki,[a,b] Frank Stein,[a] Mikhail M. Savitski,[a] Joel Selkrig,[a] Athanasios Typas[a]

[a]European Molecular Biology Laboratory (EMBL), Genome Biology Unit, Heidelberg, Germany
[b]Heidelberg University, Faculty of Biosciences, Heidelberg, Germany

**ABSTRACT** The molecular architecture and function of the Gram-negative bacterial cell envelope are dictated by protein composition and localization. Proteins that localize to the inner membranes (IM) and outer membranes (OM) of Gram-negative bacteria play critical and distinct roles in cellular physiology; however, approaches to systematically interrogate their distribution across both membranes and the soluble cell fraction are lacking. Here, we employed multiplexed quantitative mass spectrometry using tandem mass tag (TMT) labeling to assess membrane protein localization in a proteome-wide fashion by separating IM and OM vesicles from exponentially growing *Escherichia coli* K-12 cells on a sucrose density gradient. The migration patterns for >1,600 proteins were classified in an unbiased manner, accurately recapitulating decades of knowledge in membrane protein localization in *E. coli*. For 559 proteins that are currently annotated as peripherally associated with the IM (G. Orfanoudaki and A. Economou, Mol Cell Proteomics 13:3674–3687, 2014, https://doi.org/10.1074/mcp.O114.041137) and that display potential for dual localization to either the IM or cytoplasm, we could allocate 110 proteins to the IM and 206 proteins to the soluble cell fraction based on their fractionation patterns. In addition, we uncovered 63 cases, in which our data disagreed with current localization annotation in protein databases. For 42 of these cases, we were able to find supportive evidence for our localization findings in the literature. We anticipate that our systems-level analysis of the *E. coli* membrane proteome will serve as a useful reference data set to query membrane protein localization, as well as to provide a novel methodology to rapidly and systematically map membrane protein localization in more poorly characterized Gram-negative species.

**IMPORTANCE** Current knowledge of protein localization, particularly outer membrane proteins, is highly dependent on bioinformatic predictions. To date, no systematic experimental studies have directly compared protein localization spanning the inner and outer membranes of *E. coli*. By combining sucrose density gradient fractionation of inner membrane (IM) and outer membrane (OM) proteins with multiplex quantitative proteomics, we systematically quantified localization patterns for >1,600 proteins, providing high-confidence localization annotations for 1,368 proteins. Of these proteins, we resolve the predominant localization of 316 proteins that currently have dual annotation (cytoplasmic and IM) in protein databases and identify new annotations for 42 additional proteins. Overall, we present a novel quantitative methodology to systematically map membrane proteins in Gram-negative bacteria and use it to unravel the biological complexity of the membrane proteome architecture in *E. coli*.

This article followed an open peer review process. The review history can be read here.

Address correspondence to Joel Selkrig, selkrig@embl.de, or Athanasios Typas, typas@embl.de.

The inner membranes (IM) and outer membranes (OM) of Gram-negative bacteria carry out fundamental cellular functions crucial for cell viability (1). The OM directly interfaces with the extracellular environment and provides a formidable physical barrier that excludes the passage of large and hydrophobic compounds (2–4). The IM plays a vital role in ensuring selective transport of small compounds into and out of the cell,

producing and conserving energy, as well as sensing external cues and transducing information to adaptive transcriptional responses (5–7). Both membranes facilitate protein translocation from the cytoplasm to the cell envelope and/or the extracellular milieu by dedicated protein machines. Proteins targeted to the IM and OM possess distinct biochemical properties and play fundamental roles in building and maintaining cell envelope integrity. This includes correct assembly of the peptidoglycan layer, which gives bacteria their cell shape and together with the OM defines their mechanical strength (8, 9). Understanding to which membrane proteins are targeted provides important insight on the physical location of their biological activity, which is particularly useful for interrogating proteins of unknown function and investigating modular envelope protein complexes (10).

Proteins localizing to the IM or OM can be subcategorized on the basis of their distinct biophysical properties. IM proteins typically contain $\alpha$-helical transmembrane domains that mediate their integration into the phospholipid bilayer. These proteins include small-molecule transporters (e.g., ABC transporters for metal ions and sugars) and large multiprotein complexes (e.g., SecYEG translocon and ATP synthase). IM proteins are diverse in their structure, function, and domain localization; some contain structural domains that extend into the cytoplasmic and/or periplasmic space. In the OM, there are two distinct types of proteins: outer membrane proteins (OMPs), which are mostly composed of amphipathic $\beta$-strands that form a closed $\beta$-barrel structure, and lipoproteins, which are anchored to the OM via an N-terminal lipidated cysteine and typically contain a soluble domain that extends into the periplasmic space. Note that only a small fraction of lipoproteins is retained in the IM, and lipoproteins in both the IM and OM carry out diverse functions in the cell envelope, yet they still remain a largely poorly characterized group of proteins. Although lipoproteins have been traditionally considered to face the periplasmic space, some have been recently shown to reach the cell surface (11–13), even by traversing through OMPs (14–16).

While most proteins localize to, and function at, either the IM or OM, several transenvelope protein complexes span both membranes. Some of the larger complexes like flagella (17) and secretion apparatuses (18) contain dedicated IM, OM, and periplasmic components, whereas smaller ones such as the Tol-Pal complex (19–21), the PBP1a/b-LpoA/B peptidoglycan synthase complexes (22–24), the efflux pump AcrAB-TolC (25), the translocation and assembly module (TAM) (26), and all TonB-dependent transport complexes (27) possess IM or OM components that can span the envelope by reaching their interaction partner in the other membrane. Moreover, in addition to integral membrane proteins, soluble proteins can associate peripherally with membranes via stable or transient protein-protein interactions with integral membrane proteins and/or with the lipid bilayer. In the case of lipoproteins, the attachment to phospholipids is covalent and part of their biosynthesis (28). As many membrane proteins reside within protein complexes that play vital roles for cell envelope integrity, it is important to obtain a blueprint of the bacterial membrane protein composition that experimentally defines which proteins are membrane associated and the membrane to which they are targeted. Furthermore, protein localization is closely linked to protein function, and therefore a key step in ascertaining the mode of action of membrane proteins.

Localization for most proteins in *Escherichia coli* and other Gram-negative bacteria can be predicted quite accurately based on sequence information. Analysis of protein N-terminal signal sequences is commonly used to predict protein localization. For example, SignalP (29) detects signal sequences that target nascent proteins for transport across the IM via the SecYEG/SecA translocon. Similarly, TATFIND identifies substrates of the Tat translocase (30). Interestingly, periplasmic proteins bear sufficiently distinguishable biophysical, biochemical, and structural characteristics compared to their cytoplasmic counterparts, so that one can accurately discern them with signal-sequencing agnostic machine-learning predictors (31). Moreover, several tools exist to predict membrane protein localization and topology, including those that predict transmembrane $\alpha$-helices and topology of IM-transmembrane proteins such as the

TMHMM server (32), as well as algorithms to predict $\beta$-barrel folding of OMPs such as PRED-TMBB (33) and BOMP (34). *E. coli* genome databases, such as STEPdb (35), EcoCyc (36), and UniProt (37), use information from such prediction tools, together with experimental evidence, to assign protein localization. However, the difficulties of assigning protein localization solely based on structure and signal sequence prediction can lead to misannotations, as has been discussed previously (35, 38). Thus, as much of the current knowledge is based on prediction tools, it is of key importance to experimentally verify protein localization to clarify the predictive accuracy of the above-mentioned *in silico* approaches.

Proteomic-based studies of bacterial cell membranes in the past have focused on either the IM or OM proteomes separately (39–42), which precludes definitive statements about the protein allocation across the two membranes and is prone to contamination from abundant soluble proteins. To systematically examine membrane protein localization in an unbiased and systematic manner, we combined sucrose gradient membrane fractionation with quantitative proteomics (43). This allowed us to validate most of the predicted protein localizations, while resolving the protein localization of a large number of proteins with previously ambiguous localization, and uncovering some proteins with unexpected membrane localization, which are currently misannotated in databases.

## RESULTS

**IM and OM proteome separation and quantification.** To systematically assess protein localization within the bacterial cell envelope, we isolated bacterial membrane proteins by harvesting *E. coli* K-12 MG1655 grown to mid-exponential phase (Fig. 1A). Total membrane vesicles were purified, followed by IM and OM vesicle separation on a sucrose density gradient as previously described (44). The sucrose gradient was then separated into 11 fractions and analyzed by immunoblotting and sodium dodecyl sulfate-polyacrylamide gel electrophoresis (SDS-PAGE). Effective IM and OM vesicle separation was verified using SecG and BamA antiserum (Fig. 1B). Fractions 2 to 11 (f02 to f11) and the total membrane (input sample prior to fractionation) were labeled with 11-plex tandem mass tag (TMT) reagents (45) and analyzed and quantified using mass spectrometry (MS) (Fig. 1A; see Fig. S1A in the supplemental material). Multiplexing of all 11 fractions into a single MS run is what enables robust comparisons between samples and increases the quantitative power of the fractionation patterns. In total, we identified 1,605 common proteins across the two biological replicates and thus proceeded to data normalization and quantification as described in Materials and Methods (see Fig. S1A and Table S1 in the supplemental material).

To assess protein abundance in each fraction, we calculated the logarithmic fold change ($\log_2$ FC) of each sucrose gradient fraction relative to the total membrane fraction for each protein. The fractionation pattern across the 10 quantified fractions (fractions 2 to 11) was examined for each protein for each replicate (Table S1). The quantitative MS-based fractionation pattern matched immunoblot data for the control proteins, SecG for the IM and BamA for the OM (Fig. 1B, bottom). Replicate correlation between the two independent experiments for all $\log_2$ FC values was high ($R = 0.77$; Pearson correlation, $P < 0.001$; Fig. S1B). Proteome coverage was analyzed by comparing to protein localization annotations found in the curated STEPdb database (35) using UniProt identifiers (IDs) (summarized and modified as in Table S2). For membrane protein annotation categories, we had an overall 56% coverage (973 out of the total 1,741 membrane proteins annotated in the STEPdb database for *E. coli*) with several categories reaching 70% coverage, whereas nonmembrane protein categories did not exceed 25% coverage (Fig. 1C). Undetected membrane proteins may be due to lack of expression in our conditions (mid-log, LB, 37°C), generally low protein abundance, small protein size, or MS detection limitations (e.g., due to the peptide ionization properties or size after tryptic digestion). Altogether we could quantitatively assess the sucrose gradient fractionation of 1,605 proteins (Table S3), covering most membrane-annotated proteins in STEPdb.

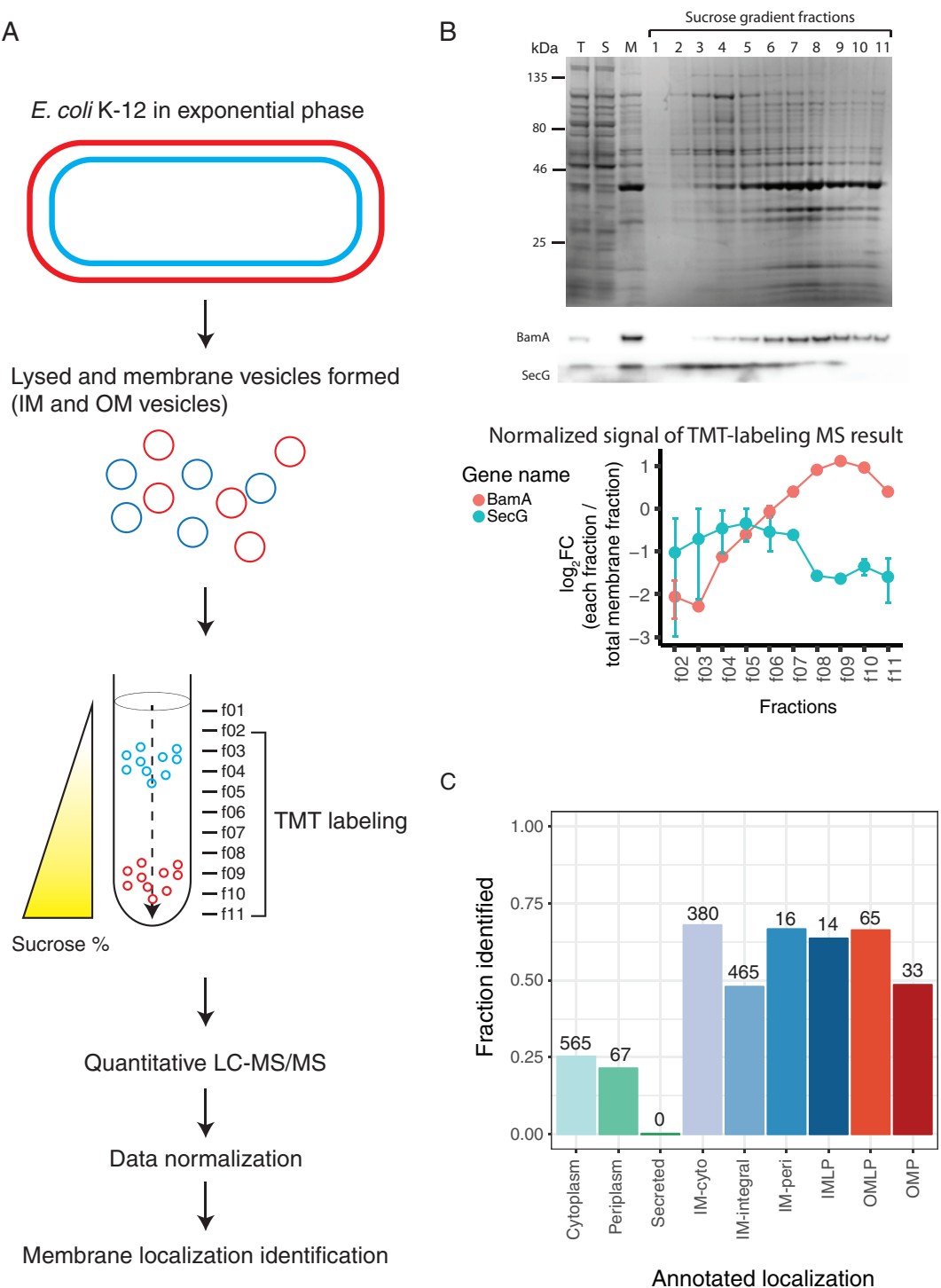

**FIG 1** Gram-negative bacterial inner and outer membrane fractionation quality control and membrane proteome coverage. (A) Schematic illustration of the method. *E. coli* cells were harvested in exponential phase ($OD_{578}$ of ~0.7), lysed, and ultracentrifuged to collect the membrane fraction containing both outer membrane (OM) (red) and inner membrane (IM) (blue) vesicles. Total membrane vesicles were separated on a sucrose density gradient, which separates IM from OM vesicles based on their distinct buoyant densities. Samples were collected into 11 fractions (f01 to f11) where f02 to f11 and total membrane sample prior to fractionation were TMT labeled and analyzed by LC-MS/MS. After data normalization (see Fig. S1 in the supplemental material), membrane localization of proteins was defined. (B) SDS-PAGE analysis of sucrose gradient fractionation. T, total cell lysate; S, soluble fraction upon ultracentrifugation; M, total membrane fraction prior to sucrose gradient fractionation. (Top) Coomassie blue-stained gel. (Middle) Immunoblot analysis for control proteins: BamA for OM, SecG for IM. (Bottom) TMT-labeling MS quantification result of the two control proteins (BamA and SecG). The graph shows the $log_2$ fold change of each fraction/total membrane fraction for fractions (f02 to f11) for BamA and SecG. Mean with standard deviation (error bars) values are plotted from the two biological replicates. (C) Fraction of proteins identified for each STEPdb localization category. Localization annotation is summarized in Table S2.

**Systematic assignment of membrane protein localization.** Sucrose density gradients are conventionally analyzed by immunoblotting to compare the abundance of a given protein within a high- or low-sucrose-density fraction (Fig. 1A and B). To systematically analyze protein localization, we used the combined averages of the high- and low-sucrose fractions ($log_2$ of f08, f09, and f10 for the high-sucrose fraction and $log_2$ of f02, f03, and f04 for the low-sucrose fraction) and calculated the difference between the two $log_2$ averages, which we referred to as the "sucrose gradient ratio" (Table S3). High values indicate a greater abundance within higher-sucrose-density fractions, as expected for OM proteins. The reverse is true for IM proteins, which exhibit low values due to their enrichment within the low-sucrose-density fractions.

To assess whether our calculated sucrose gradient ratio reflected known protein localization, we grouped these values based on localization annotation (modified from those in the STEPdb database [Table S2]). As anticipated, most IM protein categories (i.e., integral IM proteins [IM-integral], periplasmic proteins peripherally associated with IM [IM-peri], and IM lipoprotein [IMLP]) displayed a low sucrose gradient ratio, whereas the two OM protein categories, OMPs and OM lipoproteins (OMLPs), showed high sucrose gradient ratios (Fig. 2A). This striking concordance with curated annotations and our calculated localization confirms the accuracy of our methodology. We chose the 90th percentile of IM protein distribution (solid blue line) and the 10th percentile of OM protein distribution (solid red line) as cutoffs to define IM and OM protein localization using the sucrose gradient ratio (Fig. 2B), respectively. All proteins that fell between these two cutoffs were classified as soluble proteins.

Interestingly, although soluble proteins were expected contaminants in our experiments, they did not always behave as expected upon sucrose density gradient fractionation. In general, soluble proteins localized in the cytoplasm and periplasm displayed midrange sucrose gradient ratios, which is in agreement with the majority of them being contaminants and nonspecifically associated with either IM or OM vesicles. We noted that the IM-cyto category displayed bimodal characteristics with one peak being consistent with IM localization and another peak that aligned with soluble proteins (Fig. 2A and Table S3). This suggests that the IM-cyto category of proteins referred to as "peripheral IM proteins" in the STEPdb database and originally described in another study (41) consists of a mixture of proteins that have clear preferential localizations either to the cytoplasm or to the IM. We therefore did not use this category for benchmarking our data but rather kept it to later definitively allocate the primary localization of this large group of proteins. Taken together, these data show that quantitative proteomic-based analysis of sucrose gradient-fractionated membrane vesicles can rapidly and systematically localize proteins to the IM or OM.

**Unbiased clustering of sucrose gradient fractionation patterns can robustly identify protein membrane localization.** As the sucrose density-based fractionation patterns of IM and OM proteins were distinct (Fig. 1B, graph after normalization), we asked whether unbiased clustering of the fractionation patterns could be used to distinguish proteins according to their annotated membrane localization. On the basis of the results of principal-component analysis (PCA) of all identified proteins, we set the number of clusters to 4 for k-means clustering analysis (Fig. S2). This resulted in four groups containing the following number of proteins: cluster 1 with 111 proteins, cluster 2 with 650 proteins, cluster 3 with 680 proteins, and cluster 4 with 164 proteins (Fig. 3A and Table S3), where each cluster exhibited distinctive fractionation patterns (Fig. 3B). We then asked what STEPdb localization annotations are associated with proteins in the four different clusters. Proteins in cluster 1 were strongly enriched for OM proteins, cluster 2 with IM proteins, and cluster 3 and 4 with soluble proteins (Fig. 3C). Thus, k-means clustering successfully grouped proteins based on their membrane localization as a function of their sucrose gradient fractionation pattern. While each cluster was dominated by a single membrane localization annotation (OM for cluster 1, IM for cluster 2, or soluble for clusters 3 and 4), some proteins were grouped into clusters that conflicted with their STEPdb annotated localization, with the IM-cyto group featuring

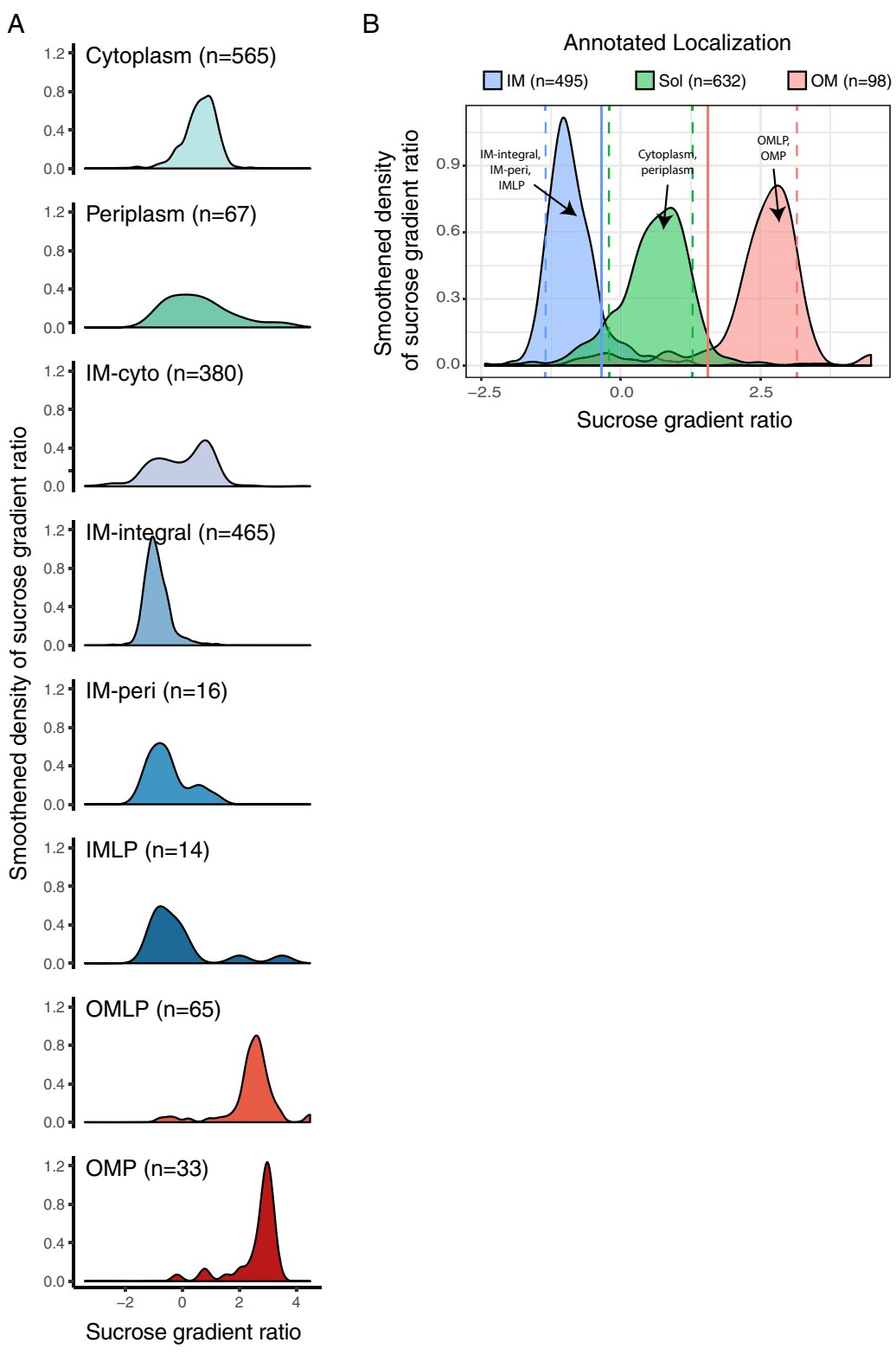

**FIG 2** Sucrose gradient ratio separates proteins according to membrane localization. (A) Smoothed distributions of sucrose gradient ratios for each protein localization category in the STEPdb database (as in Fig. 1C). Sucrose gradient ratio values were calculated as a difference of the average of high-sucrose fractions ($\log_2$ of f08, f09, and f10) and the average of the low-sucrose fractions ($\log_2$ of f02, f03, and f04) for each protein. (B) Sucrose gradient ratio of three categories (IM, OM, and soluble [Sol]) grouped from panel A. Cytoplasmic and periplasmic proteins from panel A were grouped as soluble proteins ($n = 632$), IM-integral and IM-peri as IM proteins ($n = 495$), and OM and OMLPs as OM proteins ($n = 98$). The dashed lines refer to the 10th and 90th percentiles for IM, OM, and soluble proteins for each membrane localization group. The 90th percentile for IM proteins and the 10th percentile for OM proteins are shown as solid lines, as they are used as thresholds to allocate proteins to the three categories.

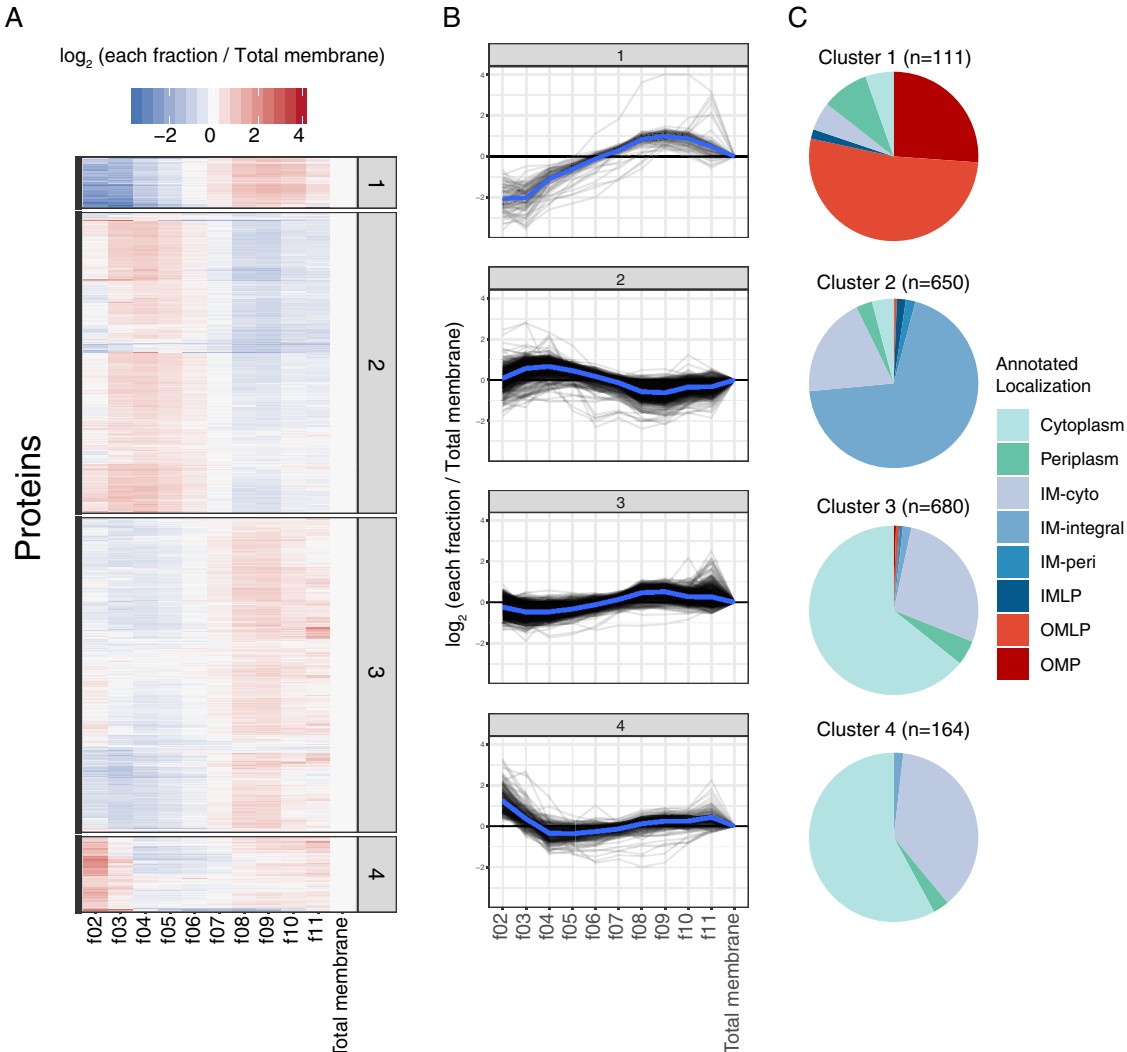

**FIG 3** Unbiased k-means clustering of sucrose gradient fractionation patterns accurately depicts protein membrane localization. The k-means clustering was based on log$_2$ fold change of each fraction to total membrane sample. (A) Heatmap representing cluster patterns. The cluster number is shown to the right of the heatmap. (B) Fractionation pattern of all proteins in each cluster (gray) and the distribution average (blue). (C) Pie chart representing annotated localization of proteins found in each cluster. The localization annotation is as shown in Fig. 1C.

prominently in various clusters (Fig. 3C). Nevertheless, these results show that systematic and unbiased clustering of sucrose gradient fractionation patterns of the membrane proteome can be used to assign membrane localization.

**Filtering and comparison of sucrose gradient ratio and clustering.** In order to increase the confidence of our calls for protein localization, we performed two further steps. First, we ran k-means clustering for the data set where the two biological replicates were treated separately. By doing this, we identified 140 (out of 1,605) proteins whose fractionation patterns between biological replicates resulted in clustering to different localization clusters (OM for cluster 1, IM for cluster 2, or soluble for clusters 3 and 4) for the two replicates (Table S3). We reasoned that this is due to irreproducibility between the two biological replicates and removed these proteins from further analysis. Second, we assessed the similarity of our two methods (k-means versus sucrose gradient ratio) in assessing protein localization. To do this, we used the thresholds of sucrose gradient ratio for IM and OM defined in Fig. 2B. We found a large overlap between the two methods for all three localization categories: IM, OM, and soluble (Fig. 4A). In total, we identified 1,368 proteins (out of 1,465 possible) with

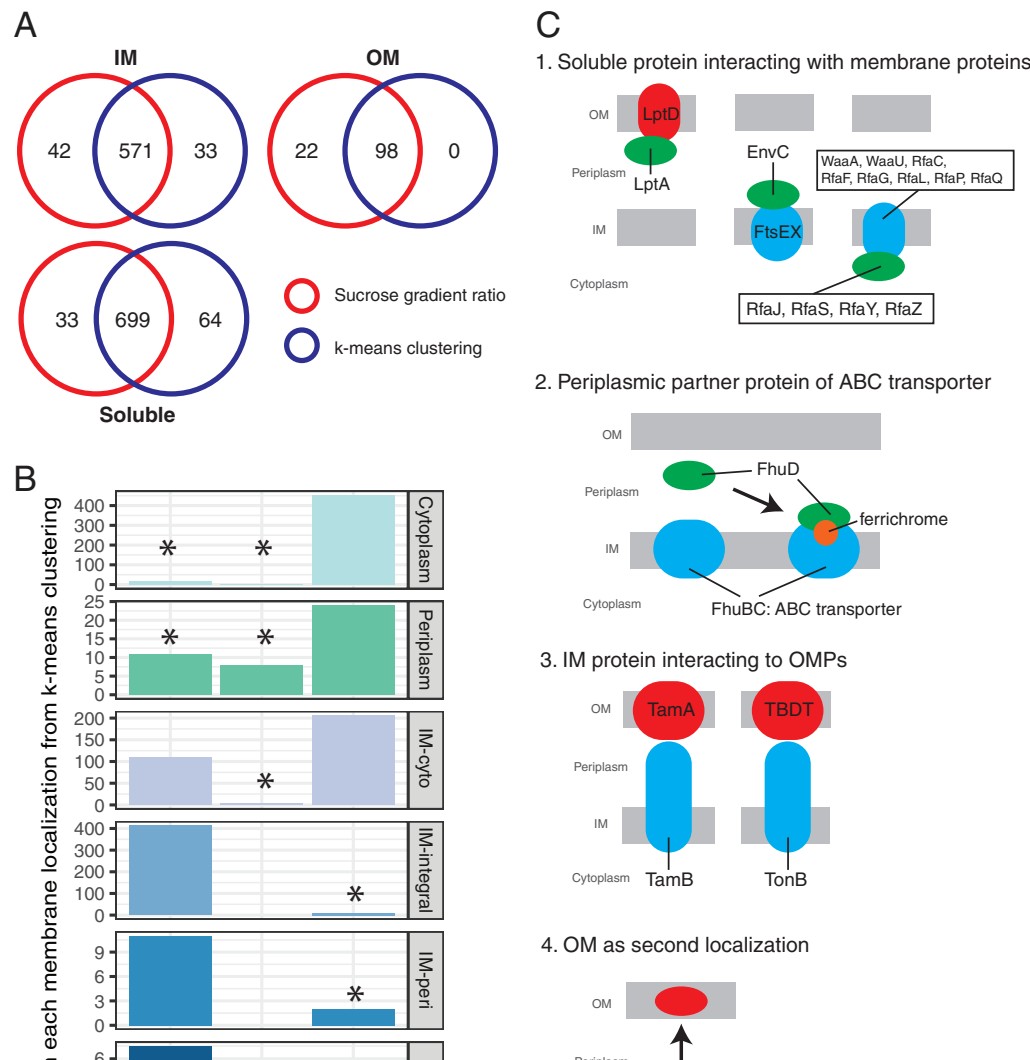

**FIG 4** Identification of potentially misannotated proteins. (A) Venn diagrams showing commonly identified proteins for each localization categories (IM, OM, and soluble), based on our two quantification methods. Protein localizations stemming from the sucrose gradient ratios are shown in red, and protein localizations from k-means clustering annotation are shown in blue. A total of 1,465 proteins were assigned localization by the different methods. The two methods agreed in the localization of 1,368 proteins, which we used as a "high-confidence" localization set (Table S3). (B) Comparison of identified localization with annotated localization for the 1,368 proteins from panel A. Protein localizations considered not to match with STEPdb annotations are marked with an asterisk. The IM-cyto category has a bimodal distribution, likely because proteins could not be confidently allocated to either the IM or cytoplasm in the STEPdb database but can be with the experimental data provided here. (C) Schematic representation explaining experimental localization of proteins not matching or partially matching the STEPdb annotation (besides IM-cyto). (Scenario 1) Soluble proteins interacting with membrane proteins. (Left) LptA interacts with OM protein, LptD/E and fractionates as OM protein as previously reported (63). (Middle) EnvC interacts with IM proteins FtsEX (64), similarly for FdoG, NapG, and RseB. (Right) Rfa proteins in cytoplasm were identified as IM proteins, possibly due to interactions with bona-fide IM Rfa proteins. (Scenario 2) Periplasmic partners of ABC transporters were detected as IM proteins presumably because they bind to the IM components and transporters when transporters are active (47). This is for example the case for FhuD interacting with

localization results that agreed between the two methods, which we further used as our high-confidence protein localization data set. When considering all 1,605 proteins, the two methods agreed for 1,456 proteins (Fig. S4A). In general, both quantification methods for protein localization worked well, and in combination provided more confident identification calls (true positive rates are 95% and 48% for overlap and nonoverlap sets, respectively; using STEPdb annotations as true positive). The clustering method worked better than the ratio cutoffs for OM proteins, but on the other hand, cluster 4 seemed to have the most inconsistent calls for protein localization (Fig. S3).

As mentioned before, IM-cyto proteins showed a bimodal sucrose gradient ratio distribution (Fig. 2A) and could be found in multiple clusters (clusters 2, 3, and 4 [Fig. 3C]). Now that we have high-confidence protein localization allocations, we could define 110 proteins as IM and 206 as soluble (Fig. 4B; see also Fig. S4B for all proteins, including lower-confidence ones). This separation is corroborated by the melting temperatures of these two group of proteins. We have previously reported that IM proteins are more thermostable than their cytoplasmic counterparts (46). In agreement with this, IM-cyto proteins categorized here to be soluble had low melting temperatures similar to those of cytoplasmic proteins (Fig. S5). In contrast, IM-cyto proteins categorized as IM proteins had higher melting temperatures, albeit not as high as integral IM proteins, presumably due to their peripheral interaction rather than integral association with the IM (Fig. S5). Thus, with the experimental conditions we tested, IM-cyto annotated proteins resulted in a mixture of soluble and IM proteins, and by using their fractionation patterns, we could allocate their predominant protein localization.

**Identification of potentially misannotated proteins.** The STEPdb database combines robust computational predictions with a wealth of experimental information to allocate protein localization in *E. coli*, and hence, we used it here as a gold standard data set to benchmark our data and decide on thresholds for making localization calls. In doing so, we noted that a small fraction of our allocations of IM, soluble, and OM proteins conflicted with their STEPdb annotations. Namely, 63 (out of 1,368) high-confidence proteins, including both membrane and soluble proteins, were found in clusters that at least partially conflicted with their corresponding STEPdb localization annotation (Table S4). We manually curated these proteins based on published literature. First, we checked the localization annotation in another recent study (38). Babu et al. (38) summarized different localization annotation databases and primary literature, including STEPdb, generated a score for protein localization, and then annotated protein localization accordingly in their study. Eleven proteins out of the 63 proteins we identified as mismatches agreed with the curated list of Babu et al. (38). Therefore, we conclude that the STEPdb localization for these 11 proteins was likely inaccurate (Table S4). We found corroborating evidence for 12 more such cases in the literature or in other prediction databases. Importantly, in these cases, rather than relying on the combined result from multiple *in silico* prediction algorithms, our data are able to provide the high-confidence experimental evidence needed to verify the localization for these proteins.

We also noted unexpected fractionation patterns for certain proteins upon sucrose density fractionation. First, we detected several proteins annotated as solely periplasmic in the STEPdb database fractionated as membrane proteins in our experiments. Many of these proteins have known interacting membrane partners, which is presumably the reason they cofractionate with either the IM (EnvC, FdoG, NapG, and RseB) or

**FIG 4** Legend (Continued)

FhuBC upon substrate (ferrichrome [orange]) binding. (Scenario 3) Transenvelope protein interaction is known. TamB spans the envelope and interacts with OM-located TamA (26) and TonB interacts with OM-located TonB-dependent transporters, resulting in a misleading sucrose gradient fractionation (distributed equally in all fractions, thus appearing as a soluble protein), which in the case of TonB has been previously shown (48, 49). (Scenario 4) Previously shown dual localization of cytoplasmic protein in OM for Dps and SeqA (65–67). A complete list of 63 proteins not matching with the STEPdb annotation, with 51 proteins explained can be found in Table S4.

mSystems®

the OM (LptA) (Fig. 4C and Table S4). The situation was similar for periplasmic components of IM ABC transporter complexes (FhuD, PstS, and SapA) which cofractionated with IM proteins, possibly as a consequence of a direct conditional association with IM proteins upon active transport (47), but were only annotated as periplasmic in STEPdb (Table S4). In total, there were 19 cases for which STEPdb had incomplete annotations.

Conversely, FecB, a known periplasmic component of an ABC transporter complex, is annotated both as IM-peri (peripherally associated with IM) and periplasmic in the STEPdb database but was identified only as soluble under our experimental conditions. In this case, we are failing to detect the IM association because the transporter is likely inactive under the conditions we used and the STEPdb annotation is more accurate (Table S4). Moreover, IM proteins known to form transenvelope complexes (e.g., TamB and TonB) failed to cluster as either IM or OM proteins in the fractionation experiments. Overall, we could reasonably explain 51 out of the 63 cases where STEPdb and our results disagreed, out of which we could find additional information that supports our localization call (42 proteins) or the original STEPdb annotation (9 proteins) (summarized in Table S4). Overall, these findings demonstrate that our quantitative assessment of protein localization captures accurately the *in vivo* biological state.

## DISCUSSION

We quantified the membrane proteome using TMT-labeling MS, which allowed us to experimentally identify localization in a systematic and unbiased manner for the majority of membrane proteins in *E. coli*. We verified current knowledge of membrane protein localization for proteins that was determined experimentally and/or predicted bioinformatically. The advantage of this method is that instead of assessing membrane protein localization via conventional immunoblotting of sucrose density gradient fractions, quantitative proteomic approaches can be used to rapidly and quantitatively assess protein localization in an antibody-independent manner.

Comparison of our data with the curated STEPdb annotation revealed high concordance. In addition, our data provided a predominant location for a large part of the *E. coli* membrane proteome referred to as peripherally associated membrane proteins (41). The STEPdb database categorizes proteins that peripherally interact with the cytoplasmic face of the IM as "peripheral IM proteins," which we referred to here for simplicity as IM-cyto (see Table S2 in the supplemental material). Although we detect the majority of IM-cyto proteins (68% of 559 proteins) in the membrane fraction (which is depleted of soluble proteins), most of them are reproducibly assigned as soluble proteins according to their sucrose fractionation pattern (206 out of 316 high-confidence calls). This absence of cofractionation with the IM proteome suggests that many of these proteins are mainly cytoplasmic in exponentially growing cells in LB, and their previous identification in membrane protein fractions in this study and others is likely because they are recurrent contaminants. We cannot exclude the possibility that some of these proteins have a conditional, low-affinity, or transient association with the IM and proteins therein or that a small fraction of the total protein amount may, at any given point, be associated with the IM. In contrast, about one third of the IM-cyto proteins exhibited clear IM fractionation patterns and thus can be confidently assigned as IM-associated proteins.

We found 63 proteins out of 1,368 which were inconsistent with the reported localization annotation in the STEPdb database. We were able to explain 51 by additional literature data. Those proteins have a wrong or missing annotation in STEPdb (42) or their function/activity makes their sucrose gradient fractionation patterns misleading (8). In most cases, sucrose gradient fractionation failed to make the right call when the protein was spanning the envelope or presumably had dual membrane localization. It is likely that the new localization is also correct for most of 12 remaining proteins (Table S4). Thus, our data are helpful for improving protein localization, even for an organism as intensively studied as *E. coli*, which has been subjected

to a plethora of targeted and systematic studies, and researchers can benefit from carefully curated databases, such as STEPdb.

In some cases, sucrose gradient fractionation patterns were indicative of protein activity. For example, periplasmic partners of IM ABC transporters often showed sucrose gradient fractionation patterns similar to those of IM proteins (Fig. 4C), which we postulate were due to their strong interaction with the cognate IM ABC transporters in their substrate-bound state (47). In other cases, they behaved as soluble proteins, which likely reflects the inactive state of the ABC transporter. Interestingly, transenvelope spanning IM proteins, such as TamB and TonB, displayed fractionation patterns similar to those of soluble proteins. This is presumably due to strong interactions with their OM counterparts, which pull a subpopulation of the protein together with OM vesicles during ultracentrifugation. Consistent with our observations, TonB was previously found in the OM fraction upon sucrose gradient fractionation (48, 49). This suggests that transenvelope IM proteins can present distinctive properties upon sucrose gradient fractionation. More broadly, it implies that sucrose gradient fractionation can provide insights into the activity and mechanical strength of specific envelope complexes during different growth stages and conditions.

Not only does this work provide a resource of sucrose gradient fractionation for 1,605 proteins, with 1,368 proteins having their cellular localization confidently assigned, the method we present can be used for rapid and systematic characterization of membrane proteomes in different contexts—growth conditions and stages, and under different cellular perturbations. Many membrane proteins are only conditionally expressed (46), whereas other proteins conditionally relocate in and out of membranes (50–52). Importantly, our method allows for the systematic mapping of membrane proteomes from other Gram-negative species for which protein localization annotation and transport mechanisms are less (if at all) studied.

## MATERIALS AND METHODS

**Bacterial culturing.** The wild-type strain used in this study is *Escherichia coli* K-12 MG1655 Δ(*argF-lac*)U169 *rprA::lacZ* (53). Bacterial cells were grown for 4 generations in LB-Lennox (referred to as LB herein) medium at 37°C with vigorous shaking at 200 rpm and collected for fractionation while still in the exponential growth phase at an optical density at 578 nm ($OD_{578}$) of 0.6 to 0.8.

**Membrane vesicle isolation and sucrose density fractionation.** Membrane vesicles were isolated and fractionated essentially as previously described (44) with the following deviations. Phosphate-buffered saline (PBS) was used as the base buffer instead of Tris. After sucrose gradient separation, 1-ml fractions were collected stepwise from the top of the gradient, yielding 11 fractionated samples that were analyzed by Coomassie blue staining and Western blotting using sodium dodecyl sulfate (SDS)-polyacrylamide gels as described below. Fractions 2 to 11 (f02 to f11), as well as an aliquot of the total input membrane sample (diluted 10 times in $H_2O$), were labeled with 11-plex tandem mass tag (TMT) and subjected to liquid chromatography coupled to tandem mass spectrometry (LC-MS/MS).

**Sample preparation and TMT labeling.** Before sample preparation for MS, proteins were solubilized by adding SDS to the samples (final concentration of 1% SDS). Samples were then sonicated for 5 min in an ultrasonic bath, heated for 10 min to 80°C, and sonicated again for another 5 min. Disulfide bonds were reduced by incubating at 56°C for 30 min in 10 mM dithiothreitol (DTT) buffered with 50 mM HEPES (pH 8.5). Reduced cysteines were alkylated by incubating for 30 min at room temperature in the dark with 20 mM 2-chloroacetamide in 50 mM HEPES buffer (pH 8.5). Samples were prepared for MS using the SP3 protocol (54, 55). On bead trypsin (sequencing grade, Promega, V5111) digestion was performed to an enzyme/protein ratio of 1:50 and incubated overnight at 37°C. Digested peptides were then recovered in HEPES buffer by collecting the supernatant after magnet-based separation from the SP3 bead, and combining the second elution wash of beads with HEPES buffer. Collected peptides were labeled with TMT10plex isobaric label reagent (ThermoFisher) (45) and with 131C label (ThermoFisher) according to the manufacturer's instructions as described below. In brief, 0.8 mg of the TMT reagents was dissolved in 42 μl of 100% acetonitrile, and 4 μl of this stock was added to the peptide sample and incubated for 1 h at room temperature. The reaction was quenched with 5% hydroxylamine for 15 min at room temperature. Then the 10 samples labeled with unique TMT10plex labels were combined into one sample. The combined sample was then cleaned up using OASIS HLB μElution plater (Waters). The samples were separated through an offline high-pH reverse-phase fractionation on an Agilent 1200 Infinity high-performance liquid chromatography system which was equipped with a Gemini $C_{18}$ column (3 μm; 110 Å; 100 by 1.0 mm; Phenomenex). The fractionation was performed as previously described (56). Samples were pooled into a total of 12 fractions.

**Mass spectrometry data acquisition.** Chromatography was performed using an UltiMate 3000 RSLC nano LC system (Dionex) fitted with a trapping cartridge (μ-Precolumn $C_{18}$ PepMap 100; 5 μm; 300-μm inner diameter [i.d.] by 5 mm; 100 Å) and an analytical column (nanoEase M/Z HSS T3 column 75 μm by

250 mm C$_{18}$; 1.8 $\mu$m; 100 Å; Waters). Trapping was carried out with a constant flow of solvent A (0.1% formic acid in water) at 30 $\mu$l/min onto the trapping column for 6 min. Subsequently, peptides were eluted via the analytical column with a constant flow of 0.3 $\mu$l/min with an increasing percentage of solvent B (0.1% formic acid in acetonitrile) from 2% to 4% in 6 min, from 4% to 8% in 1 min, then 8% to 25% for a further 71 min, and finally from 25% to 40% in another 5 min. The outlet of the analytical column was coupled directly to a Fusion Lumos (Thermo) mass spectrometer using the proxeon nanoflow source in positive ion mode.

Peptides were introduced into the Fusion Lumos via a Pico-Tip Emitter (360-$\mu$m outer diameter [o.d.] and 20-$\mu$m i.d.; 10 $\mu$m tip [New Objective]) and an applied spray voltage of 2.4 kV. The capillary temperature was set at 275°C. A full mass scan was acquired with a mass range of 375 to 1,500 $m/z$ in profile mode in the Orbitrap with resolution of 120,000. The filling time was set at maximum of 50 ms with a limitation of $4 \times 10^5$ ions. Data-dependent acquisition (DDA) was performed with the resolution of the Orbitrap set at 30,000, with a fill time of 94 ms and a limitation of $1 \times 10^5$ ions. A normalized collision energy of 38 was applied. MS$^2$ data were acquired in profile mode.

**MS data analysis.** IsobarQuant (57) and Mascot (v2.2.07) (58) were used to process the acquired data, which was then searched against a UniProt *Escherichia coli* proteome database (UniProt accession no. UP000000625; downloaded on 14 May 2016) containing common contaminants and reversed sequences (37). The following modifications were included into the search parameters: Carbamidomethyl (C) and TMT10 (K) (fixed modification), Acetyl (Protein N-term), Oxidation (M) and TMT10 (N-term) (variable modifications).

For the full scan (MS1), a mass error tolerance of 10 ppm, and for MS/MS (MS2) spectra of 0.02 Da was set. Further parameters were set: trypsin as protease with an allowance of maximum two missed cleavages; a minimum peptide length of seven amino acids; at least two unique peptides were required for a protein identification. The false-discovery rate on the peptide and protein levels was set at 0.01.

**Statistical analysis of MS data.** The protein.txt output files of IsobarQuant (57) were processed using the R programming language (59). As a quality criterion, only proteins that were quantified with at least two unique peptides were used. Raw tmt reporter ion signals (signal_sum columns) were first batch cleaned using the removeBatchEffect function from limma (60) and then normalized using the vsn package (61). Normalized data were clustered in four clusters using the kmeans function of the stat package in R.

**SDS-PAGE and Coomassie blue staining.** Protein samples were solubilized and reduced by boiling at 95°C for 5 min in Laemmli loading buffer (200 mM Tris-HCl [pH 6.8], 8% SDS, 40% glycerol, 400 mM DTT, 0.02% bromophenol blue). Solubilized samples were loaded and separated in gradient gels of 4 to 20% acrylamide (Teo-Tricine gels from Expedeon, NXG42012) using the running buffer (Run-Blue running buffer consisted of 0.8 M Tricine, 1.2 M triethanolamine, and 2% SDS). Bio-Rad systems were used, applying 100 V per chamber. For Coomassie blue staining, gels were incubated in staining solution (50% methanol, 40% H$_2$O, 10% acetic acid, 1 g Brilliant Blue R250 per 1 liter) for 1 h, and destained with destaining solution (40% ethanol, 10% acetic acid, 50% H$_2$O) until the desirable signal was achieved. Incubations were performed at room temperature with constant moderate mixing by rocking.

**Immunoblot analysis.** Proteins were separated on acrylamide gels as described above and transferred to methanol-activated polyvinylidene difluoride (PVDF) membranes (Merck, IPVH00010), using Western blot transfer buffer (3.03 g Tris, 14.4 g glycine, 200 ml methanol [all per 1 liter]) for 1.5 h at 100 V. All the incubation steps from here on were performed with constant moderate agitation by using rocking platforms. Membranes were blocked for 1 h with 5% skim milk in Tris-buffered saline with Tween 20 (TBST) (20 mM Tris, 10 mM NaCl, 0.1% Tween 20) and then incubated with appropriately diluted primary antibodies (anti-BamA diluted 1:10,000 and anti-SecG diluted 1:6,000) in 5% skim milk in TBST overnight at 4°C. After three 5-min washes with TBST, membranes were incubated for 1 h with secondary anti-rabbit antibodies conjugated with horseradish peroxidase (HRP) (catalog no. NA934; GE Health Care) diluted 1:10,000 in 5% skim milk in TBST. After these antibody incubations, membranes were washed again three times for 5 min with TBST. Proteins were detected by adding ECL substrate (catalog no. RPN2106; GE Healthcare), then exposing and visualizing using a digital developing machine (ChemiDoc touch imaging system).

**Databases.** UniProt (37) was used as the source for protein ID and sequences. Protein localization information on the STEPdb database (35) was summarized and modified as in Table S2 in the supplemental material. Modification includes assignment of single localization for proteins with two or more localization annotations.

**Data availability.** The mass spectrometry proteomic data have been deposited to the ProteomeXchange Consortium via the PRIDE (62) partner repository with the data set identifier PXD016403. The code and pipelines used for data analysis are available upon request.

## SUPPLEMENTAL MATERIAL

Supplemental material is available online only.

**FIG S1**, PDF file, 0.6 MB.

**FIG S2**, PDF file, 1.1 MB.

**FIG S3**, EPS file, 1.6 MB.

**FIG S4**, EPS file, 1.7 MB.

**FIG S5**, EPS file, 1.8 MB.

**TABLE S1**, XLSX file, 1.4 MB.

**TABLE S2**, XLSX file, 0.1 MB.
**TABLE S3**, XLSX file, 0.4 MB.
**TABLE S4**, XLSX file, 0.02 MB.

## ACKNOWLEDGMENTS

We thank Tassos Economou for critically reading the manuscript and providing feedback; members of Typas lab for valuable discussions; H. Tokuda for the SecG antibody; T. Lithgow for the BamA antibody; and members of the EMBL Proteomics Core Facility (PCF), especially Mandy Rettel and Dominic Helm, for assisting with sample preparation and data acquisition.

We acknowledge funding from EMBL for this research. J.S. was supported by fellowships from the EMBL Interdisciplinary Postdoc (EIPOD) program under Marie Skłodowska-Curie Actions COFUND (grant 291772). A.S. was supported by the DFG under a grant in the priority program SPP1617 and EMBL.

This research was conducted by A. Sueki as part of a collaboration for a joint Ph.D. degree from EMBL and Heidelberg University.

We declare that we have no conflicts of interest.

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
