## [Reviewer comments · mSystems]

Systematic localization of *Escherichia coli* membrane proteins

Anna Sueki, Frank Stein, Mikhail Savitski, Joel Selkrig, and Athanasios Typas

Corresponding Author(s): Athanasios Typas, European Molecular Biology Laboratory

Review Timeline:

Submission Date:	November 27, 2019
Editorial Decision:	January 29, 2020
Revision Received:	February 12, 2020
Accepted:	February 14, 2020

Editor: Caroline Ajo-Franklin

Reviewer(s): The reviewers have opted to remain anonymous.

Transaction Report:

DOI: <https://doi.org/10.1128/mSystems.00808-19>

January 29, 2020

Dr. Athanasios Typas
European Molecular Biology Laboratory
Genome Biology Unit
Meyerhofstrasse 1
Heidelberg
Germany

Re: mSystems00808-19 (Systematic localization of Gram-negative bacterial membrane proteins)

Dear Dr. Athanasios Typas:

We have received the reviews of your manuscript and they are enclosed with this letter. The reviewers have raised points that require response and minor revision of the manuscript before it is suitable for publication. In particular, we request that you revise the title to be more specific to the findings of your study. However, with response and minor revisions, the manuscript should be acceptable for publication in mSystems.

Below you will find the comments of the reviewers.

To submit your modified manuscript, log onto the eJP submission site at <https://msystems.msubmit.net/cgi-bin/main.plex>. If you cannot remember your password, click the "Can't remember your password?" link and follow the instructions on the screen. Go to Author Tasks and click the appropriate manuscript title to begin the resubmission process. The information that you entered when you first submitted the paper will be displayed. Please update the information as necessary. Provide (1) point-by-point responses to the issues raised by the reviewers as file type "Response to Reviewers," not in your cover letter, and (2) a PDF file that indicates the changes from the original submission (by highlighting or underlining the changes) as file type "Marked Up Manuscript - For Review Only."

Please return the manuscript within 60 days; if you cannot complete the modification within this time period, please contact me. If you do not wish to modify the manuscript and prefer to submit it to another journal, please notify me of your decision immediately so that the manuscript may be formally withdrawn from consideration by mSystems.

To avoid unnecessary delay in publication should your modified manuscript be accepted, it is important that all elements you upload meet the technical requirements for production. I strongly recommend that you check your digital images using the Rapid Inspector tool at <http://rapidinspector.cadmus.com/RapidInspector/zmw/>.

Corresponding authors may join or renew ASM membership to obtain discounts on publication fees.

Need to upgrade your membership level? Please contact Customer Service at Service@asmusa.org.

Sincerely,

Caroline Ajo-Franklin

Editor, mSystems

Journals Department
Reviewer comments:

Reviewer #1 (Comments for the Author):

- 1□ Please check whether the words "Gram-negative bacterial" of the title are too broad.
- 2□ Whether the proteins isolated in mid exponential phase of the 148-149 line can represent all of the membrane proteins of *E. coli* K-12 MG1655.
- 3□ In the manuscript, the innovation of this paper was not reflected.
- 4□ The experimental treatment in the manuscript is two sets of repetitions, but on lines 245-246 "We reasoned that this is due to irreproducibility between the replicates and removed these proteins from further analysis." Please explain again after verifying again.
- 5□ Please modify the legend in Figure 3 to be consistent with the others instead of using "left, center, right" for explanation.
- 6□ Please add the ratio between the number of proteins in the study and the total number of related proteins that have been found so far.

Reviewer #2 (Comments for the Author):

In this manuscript Sueki and co-workers combine multiplex quantitative proteomics with gradient sucrose density centrifugation of inner and outer membrane proteins to provide localization annotation of the whole *Escherichia coli* envelope proteome. This systematic analysis allows the authors to validate most of the predicted *E. coli* protein localization. However, the proposed methodology seems best suited to assign localization of OM proteins. Remarkably, the authors are able to explain most of the discrepancies between the proposed membrane protein localizations and the reported annotations in the STEP database. The work is original and solid, the experiments are well planned and rigorously performed; the results well presented. The proposed methodology could be used to systematically map membrane protein localization in poorly characterized Gram-negative species.

Minor comments

Introduction

Lines 55-57: among IM vital roles energy production and conservation should be included

Lines 72-74: please include IM lipoproteins among proteins associated/targeted to the IM.

Results

Are IM and OM vesicles produced in similar proportions by E. coli cells?

Localization by sucrose gradient fractionation seems very accurate in assigning localization of proteins to OM (OMLP and OMP) but less accurate in assigning localization of peripheral IM proteins (IM cyto). Could this bias be due to different amount of inner membrane vesicles produced by E. coli cells? (compared to outer membrane vesicles). In other words, could different IM and OM vesiculation impact on the assessment of protein localization?

Figure 2 panel A:

IMLP and OMLP share very similar sucrose gradient ratio despite their localization in inner and outer membrane, respectively. Could the authors comment?

Figure 2 panels A and B

Please explain "density" parameter reported on y axis

Figure 1 legend

Line 426: please remove (middle panel), repetition

Response to reviewers

Reviewer #1 (Comments for the Author):

1. Please check whether the words "Gram-negative bacterial" of the title are too broad.

We thank the reviewer for bringing this to our attention. Indeed, the title is too broad. We have since changed the title to "Systematic localization of *Escherichia coli* membrane proteins".

2. Whether the proteins isolated in mid exponential phase of the 148-149 line can represent all of the membrane proteins of *E. coli* K-12 MG1655.

Not all, but a significant fraction; overall, we cover 973/1741 membrane proteins (line 175 and Figure 1C) – requiring at least 2 unique peptides per protein for identification. Many proteins may be only expressed in other conditions, not fly well for MS experiments, or have low overall abundance. In general, the number/fraction of detected proteins is on par with previous proteomics studies detecting membrane proteins in *E. coli*. Nevertheless, we have now specified these limitations in the text (lines 178-181).

3. In the manuscript, the innovation of this paper was not reflected.

We felt that this was clear – using quantitative MS (TMT labeling) as a readout of the sucrose gradient fractionation, which has enabled us to derive robust quantitative fractionation patterns for all proteins. This is what give us the power, and together with our data analysis pipeline have allowed us to make accurate predictions for protein localization, recapitulating years of extensive studies in one go.

We have added text in the abstract and results (lines 20, 157-160) to emphasize that the power of our approach lies on the MS quantitative readout.

4. The experimental treatment in the manuscript is two sets of repetitions, but on lines 245-246 "We reasoned that this is due to irreproducibility between the replicates and removed these proteins from further analysis." Please explain again after verifying again.

We apologize for this confusion. We meant "between the two independent biological replicates"; we changed text accordingly (line 250-254).

5. Please modify the legend in Figure 3 to be consistent with the others instead of using "left, center, right" for explanation.

We have changed Figure 3 to be as the separate panels (A, B and C) as requested.

6. Please add the ratio between the number of proteins in the study and the total number of related proteins that have been found so far.

We used the STEPdb database as the reference of membrane proteins (note that some annotated proteins therein have not been identified experimentally), and calculated the ratio of identified proteins in our study compared to that. The identified 56% is mentioned in lines 174-176 and visualized in Fig. 1C for each category of proteins. We have also modified the text for clarity - from "973 out of the 1741 proteins" to "973 out of the total 1741 membrane proteins annotated in *E. coli*".

Reviewer #2 (Comments for the Author):

In this manuscript Sueki and co-workers combine multiplex quantitative proteomics with gradient sucrose density centrifugation of inner and outer membrane proteins to provide localization annotation of the whole Escherichia coli envelope proteome. This systematic analysis allows the authors to validate most of the predicted E. coli protein localization. However, the proposed methodology seems best suited to assign localization of OM proteins. Remarkably, the authors are able to explain most of the discrepancies between the proposed membrane protein localizations and the reported annotations in the STEP database. The work is original and solid, the experiments are well planned and rigorously performed; the results well presented. The proposed methodology could be used to systematically map membrane protein localization in poorly characterized Gram-negative species.

We thank Review #2 for his/her comments on our work.

Minor comments

Introduction

Lines 55-57: among IM vital roles energy production and conservation should be included

We thank the reviewer for this suggestion, we have now included this in lines 56-57.

Lines 72-74: please include IM lipoproteins among proteins associated/targeted to the IM.

We thank the reviewer for this suggestion, we have included this in line 80-82.

Results

Are IM and OM vesicles produced in similar proportions by E. coli cells?

Theoretically one would think there are slightly more OM vesicles as the area of OM is larger than IM (we did not find any studies which accurately compares the level of vesicle production for two membranes). However, there are more proteins targeted to the IM, which is reflected by signal sum we see for the sucrose gradient fractions (Figure S1A): there is more signal from lower fractions (enriched in IM proteins) compared to higher sucrose fractions (enriched in OM proteins). Hence whatever difference we have on vesicle production, we isolate/detect more IM proteins in absolute number.

Localization by sucrose gradient fractionation seems very accurate in assigning localization of proteins to OM (OMLP and OMP) but less accurate in assigning localization of peripheral IM proteins (IM cyto). Could this bias be due to different amount of inner membrane vesicles produced by E. coli cells? (compared to outer membrane vesicles). In other words, could different IM and OM vesiculation impact on the assessment of protein localization?

Sucrose gradient fractionation and the observed ratios have been used for decades to define whether proteins localize to the OM or IM. In our case, we can make a call for the localization of a protein when this is identified on both biological replicates. We don't think different vesiculation plays a role (see above), but we do seem to have less power to detect both integral IM and OM proteins (Figure 1C). We assume this is due to MS technical reasons (low abundance, proteins deprived of lysines or lack of good flying peptides).

Concerning accuracy, we think we are very accurate in assigning both IM and OM proteins. The IM-cyto is a specific case, which we explain in text in detail (lines 270-284 and 342-358). We think that the reported IM association for IM-cyto is for most cases wrong or due to weak/conditional interactions. As we explain, this is corroborated by the melting behavior of these proteins (Figure S5).

More generally, for all soluble proteins, cytoplasmic or periplasmic, there is the chance that they have conditional or weaker interaction with the membrane next to them (or integral proteins in the membrane). As cytoplasmic proteins are an order of magnitude more than periplasmic, and IM can be reached by both cytoplasmic and periplasmic proteins, this makes IM peripherally associated proteins a much larger group than the OM peripherally associated proteins – hence more questionable annotations in literature. As the reviewer may appreciate, periplasmic proteins and IM-peri are the other “problematic groups” due to a similar potential for peripheral association to membranes (Figure 4B). We try to explain these differences and make the call about the accurate annotation for each different annotation in Table S4 and a separate results section (lines 286-330). In the majority of the cases where there is difference between our data and databases, this is due to incomplete or mis-annotation in the databases.

Figure 2 panel A:

IMLP and OMLP share very similar sucrose gradient ratio despite their localization in inner and outer membrane, respectively. Could the authors comment?

We suspect the legend alignment was misleading. We apologize for this; we have now increased the separation between the protein categories. IMLP (third from the bottom) and OMLP (second from the bottom) shows distinctive sucrose gradient ratio coverage. Same is to be seen clearly also in Figure S4B.

Figure 2 panels A and B

Please explain "density" parameter reported on y axis

We renamed the y-axis in Figure 2 to “smoothened density of sucrose gradient ratio”.

Figure 1 legend

Line 426: please remove (middle panel), repetition

Done

February 14, 2020

Dr. Athanasios Typas
European Molecular Biology Laboratory
Genome Biology Unit
Meyerhofstrasse 1
Heidelberg
Germany

Re: mSystems00808-19R1 (Systematic localization of *Escherichia coli* membrane proteins)

Dear Dr. Athanasios Typas:

Your manuscript has been accepted, and I am forwarding it to the ASM Journals Department for publication. For your reference, ASM Journals' address is given below. Before it can be scheduled for publication, your manuscript will be checked by the mSystems senior production editor, Ellie Ghatineh, to make sure that all elements meet the technical requirements for publication. She will contact you if anything needs to be revised before copyediting and production can begin. Otherwise, you will be notified when your proofs are ready to be viewed.

Sincerely,

Caroline Ajo-Franklin
Editor, mSystems

Journals Department
Supplemental Material: Accept
Supplementary Figure 2: Accept
Supplemental Material: Accept
Supplementary Figure 1: Accept
Supplementary Figure 4: Accept
Supplementary Figure 5: Accept
Supplementary Figure 3: Accept
Supplemental Material: Accept
Supplemental Material: Accept